# Cardiac Surgery in Nonagenarians Following the TAVI/TMVI Era: A Multicenter 23-Year Comparative Analysis [note 1]

**DOI:** 10.3390/jcm12062177

**Published:** 2023-03-10

**Authors:** Giuseppe Nasso, Giuseppe Santarpino, Nicola Di Bari, Khalil Fattouch, Ignazio Condello, Marco Moscarelli, Mauro Del Giglio, Domenico Paparella, Mauro Lamarra, Carlo Savini, Roberto Coppola, Vinicio Fiorani, Giuseppe Speziale

**Affiliations:** 1Department of Cardiac Surgery, Anthea Hospital, GVM Care & Research, 70100 Bari, Italy; 2Department of Cardiac Surgery, Città di Lecce Hospital, GVM Care & Research, 73100 Lecce, Italy; 3Department of Cardiac Surgery, Paracelsus Medical University, 40100 Nuremberg, Germany; 4Department of Experimental and Clinical Medicine, “Magna Graecia” University, 88100 Catanzaro, Italy; 5Department of Cardiac Surgery, “Aldo Moro” University, 70100 Bari, Italy; 6Department of Cardiac Surgery, Maria Eleonora Hospital, GVM Care & Research, 90121 Palermo, Italy; 7Department of Cardiac Surgery, Maria Pia Hospital, GVM Care & Research, 10024 Torino, Italy; 8Department of Cardiac Surgery, Santa Maria Hospital, GVM Care & Research, 70100 Bari, Italy; 9Department of Cardiac Surgery, University of Foggia, 71121 Foggia, Italy; 10Department of Cardiac Surgery, Maria Cecilia Hospital, GVM Care & Research, 40100 Cotignola, Italy; 11Department of Cardiac Surgery, Villa Torri Hospital, GVM Care & Research, 40100 Bologna, Italy; 12Department of Cardiac Surgery, ICLAS Hospital, GVM Care & Research, 16135 Rapallo, Italy; 13Department of Cardiac Surgery, Salus Hospital, GVM Care & Research, 40100 Reggio Emilia, Italy; 14Department of Cardiac Surgery, San Carlo di Nancy Hospital, GVM Care & Research, 00042 Rome, Italy

**Keywords:** elderly, nonagenarians, cardiac surgery, heart team

## Abstract

Background: Studies reporting on the outcome of 90-year-old patients undergoing cardiac surgery are scant in literature; and currently, those regarding the implementation of trans-catheter techniques number even fewer. Methods: We compared patients aged >89 years operated on between 1998 and 2008 at 8 Italian cardiac surgery centers, with patients of the same age operated on between 2009 and 2021. All of the patients were operated on with “open” surgery, with the exclusion of percutaneous valve repair/implantation procedures. Results: The patients of the two groups (group 98-08—127 patients, and group 09-21—101 patients) had comparable preoperative risk factors in terms of the LogEuroSCORE (98-08: 21.3 ± 6.1 vs. 09-21: 20.9 ± 11.1, *p* = 0.12). There was a considerable difference in the type of surgery (isolated valve, isolated coronary, and combined surgery, 46.5, 38.5, and 15% vs. 52, 13, and 35% in 98-08 and 09-21, respectively, *p* = 0.01). Analogous operating durations were recorded (cross-clamp time: 98-08: 46 ± 28 min vs. 09-21: 51 ± 28 min, *p* = 0.06). The number of packed bypasses was lower in 09-21 (1.3 ± 0.6 vs. 2.4 ± 1.2, *p* = 0.001). In the postoperative period, there was a statistically significant difference in the 30-day survival in favor of the “more recent” patients (98-08: 17 deaths (13.4%) versus 09-21: 6(5.9%); *p* = 0.001), also confirmed in the subgroups (12.2% vs. 0% in isolated coronary surgery, *p* < 0.001; and 12.3% vs. 0% in isolated valve surgery, *p* < 0.001). Conclusions: Accurate pre-, intra-, and post-operative evaluation/management to reduce biological impacts facilitate significant improvements in the outcomes in nonagenarian patients when compared to the results recorded in previous years.

## 1. Introduction

The increase in the life expectancy of the population, particularly in the Western World, can be owed to the development of surgical and anesthesiological techniques [1]; this also enables surgery in elderly patients to be more efficient by the day. However, the number of patients with cardiovascular disease, including those with an indication for cardiac surgery, is also rising. Specifically, for the category of very elderly patients, various parts of the world have reported cardiac surgery performed on 90-year-old patients with acceptable results, as judged by the authors themselves in terms of morbidity and mortality, particularly for elective procedures performed early at the onset of the patient’s symptoms [2,3,4,5]. Regardless, the ambiguity in terms of the correct preoperative risk assessment remains, when considering all the possible alternative interventional procedures that could be performed [6]. Therefore, probing whether technical and technological advances and collective strategic choices have led to a possible improvement in the outcome of these patients is imperative [7]. Our primary objective, which began by collecting data on such patients from the end of the 1990s [2], is to precisely compare 90-year-old patients operated on in an era that we could define as “pre TAVI and TMVR”, which is not limited to patients exclusively suffering from valve pathology, to those who are now chosen to undergo cardiac surgery following collective analysis by a “heart team” that deems the suitability of an alternative strategy for individual patients.

## 2. Patients and Methods

The study was conducted on two population sets. The first is made up of patients aged 90 years or over who were operated on between the years 1998 and 2008 at 8 Italian cardiac surgery centers of the Villa Maria Group (GVM), whose hospital results have been published previously [2]. The data from the patients were amassed using a unique electronic database of the GVM group per institutional protocol. The data regarding the patients of the second group were obtained using the same electronic database, i.e., patients of the same age but operated on from 2009 to 2021, with the same heart surgeries, within the same hospital group, to which 3 cardiac surgery centers were added in the course of the past decade, totaling 11 cardiac surgery centers. All patients operated on at the Villa Maria group, are carefully assessed for preoperative risk, evaluating them from “low risk” to “inoperable”. Given their advanced age, patients in the 98-08 group were also routinely assessed using the Duke Activity Status Index (DASI) [8,9]. Additionally, informed patient care and shared decision-making play a significant role in analyzing the surgical indication for individuals. Subsequently, for the 09-21 group, the patients were evaluated using the DASI and given possible endovascular alternatives for valve surgery. By virtue of the continuous advancement of the guidelines in the field of valve and coronary surgery, the patients were increasingly evaluated for surgical operability, through the establishment of multidisciplinary teams, namely “heart teams”, in all the hospitals mentioned above. The variables included all patients considered operable and operated on with “open” cardiac surgery (“on” or “off” pump for coronary surgery; conventional or minimally invasive for valve surgery) in the Euroscore predicted risk system and excluded percutaneous valve implant procedures or percutaneous mitral valve procedures. Furthermore, the hospital outcomes and mortality up to 30 days were analyzed and compared between the two groups. The study protocol was approved by the institutional review board. Given the retrospective nature of the study, since all the patient data were treated anonymously, and no additional diagnostic or therapeutic procedures were conducted on patients, individual informed consent was not deemed necessary.

### Statistical Analysis

The data were analyzed using SPSS (Statistical Package for Social Sciences, SPSS Inc., Chicago, IL, USA) software version 11.0 for Windows. Patients were divided into 2 groups according to the date of operation performed: group 98-08 (if operated between 1998 and 2008); and group 09-21 (if operated between 2009 and 2021). A Shapiro-Walk test was performed for determining normality. Continuous variables were presented as mean ± SD and categorical variables as percentages. Group comparisons were done using χ^2^ test for categorical variables and 1-way ANOVA for continuous variables. Student t-test was used for continuous variables due to the data with respect to the independence, normality, and homogeneity of variances. Postoperative complications were defined as those occurring within 30 days of surgery; therefore, no time-to-event analysis was used to identify risk factors for postoperative complications.

## 3. Results

The preoperative characteristics of the 98-08 and 09-21 groups are described in Table 1 which shows that there are significant differences between the two groups in the recorded variables. The 09-21 group, despite having more risk factors, has a similar mean predicted risk to the 98-08 group presented. Table 2 compares the patients undergoing isolated coronary surgery with patients undergoing isolated aortic valve surgery. In addition, it shows how although a higher number of patients underwent minimally invasive surgery in the 09-21 group, distinctly in the case of valve surgery, the operating times do not differ between the two groups. Moreover, the fewer patients undergoing coronary artery surgery in the 09-21 group correlated with fewer prepackaged bypasses, with a higher percentage of patients in this group undergoing a hybrid procedure with percutaneous completion. 

Table 3 compares the postoperative results between the two groups, where the 98-08 group reports significant improvement in the 30-day survival rate. We further classify and analyze these results for the subgroups of coronary isolated and isolated aortic valves.

## 4. Comments

Cardiac surgery in nonagenarians represents a perspective of the current scenario of numerous comorbidities affecting patients with cardio-vascular disease, leading to an increase in healthcare costs and the challenging cost-effectiveness ratio for the entire healthcare system. Healthcare costs could be estimated based on the duration of the average hospitalization; “classic” admission usually is defined as 24–48 h of hospital stay, whereas patients undergoing cardiac surgery reportedly spend 10–12 days in the intensive care [2,10].

Our study has shown that ninety-year-old patients operated on in the last ten years have a significantly higher 30-day survival rate than patients of the same age that we operated on twenty years ago.

There are various reasons for the substantial change in the outcomes of these patients which requires extensive analysis.

Presently, studies analyzing the outcomes of these patients have not been sufficiently reported. In fact, most studies in the literature on this topic have a very limited number of patients [2,3,4,5,10,11,12,13,14,15,16]: from just over 10 patients [3,15] to a maximum of 49 [11]. Two articles present slightly more than 100 patients [4,5]; however, in these papers, nonagenarian patients are drawn from large national databases consisting of thousands of patients. Thus, our study, with more than 200 patients aged over 90 enlisted, currently represents the largest case study from a single hospital group.

Furthermore, particularly for the “elderly” population, we must consider that in recent years during our ongoing comparative population study, there has been a revolution in the treatment of severe calcific aortic valve stenosis which corresponds to the current guidelines that place the use of transcatheteral procedures as a preferential indication in patients over 75 years of age [17] or those that are contraindicated to go through conventional surgical treatment. However, there are conditions, as also highlighted in the aforementioned guidelines [17], that can occur frequently in very elderly patients, such as a “challenging or impossible transfemoral access” or a “thrombus in aorta or left ventricle”, which indicates for a conventional surgical approach rather than TAVI, regardless of age. As far as the treatment of the mitral valve is concerned, the current guidelines [17] still favor surgical treatment, preferably reparative, except in cases of high risk or inoperability, in which case, an endovascular approach to the mitral valve is endorsed when anatomically feasible.

Another aspect to consider is the increasingly widespread introduction of minimally invasive and hybrid approaches to explain the results we recorded with our study.

The minimally invasive approach for valve surgery, both aortic and mitral, are not mentioned in the guidelines, except when implying a potential improvement in the respiratory function of the patients [17]. Yet, there are now numerous multicentric reports that demonstrate the advantages of minimally invasive surgery which is not time intensive and is suitable even for elderly patients [18]. In the past decade, patients in the 09-21 group had begun receiving valve devices during minimally invasive surgery, e.g., sutureless prostheses which are precisely aimed at reducing ischemic times, thereby benefiting more delicate patients, such as the elderly [19]. The choice of a hybrid approach, or rather of a single revascularization of the anterior wall in a patient with multivessel coronary artery disease, followed by completion with a transcatheter/coronarography approach, is a strategy proposed by many, particularly for the treatment of patients who are frail, in order to avoid extracorporeal circulation [20]; nonagenarian patients may fall into this category of patients. Contrary to this, coronary angioplasty techniques are technically feasible in ninety-year-olds, but there is an increased risk of thrombosis, cardiogenic shock, bleeding and vascular complications, myocardial infarction, reoperation, greater anatomical complexity, and complications of dual antiplatelet therapy. This suggests that the results of these 30-day percutaneous procedures in 90-year-olds are highly variable, with mortality ranging from 0 to 34%, and significantly worse than results in 80-year-old patients [21,22,23,24,25]. Furthermore, it should be noted that the strategy characterized by the combination of coronary angioplasty and TAVI has not yet been explored in all patients and even fewer in the ninety-year-old age group.

Additionally, the concept of frailty plays a vital role as all nonagenarians cannot be considered “frail”, and the surgical outcome is more closely related to the patient’s frailty than to their chronological age [26,27]. The accuracy of the STS and Euroscore I and II surgical risk assessment and prediction systems could be variable because they are not validated in the category of “rare” patients, as they fail to reach the 1.5% of the population of patients operated on within the GVM hospital group. The assessment of frailty is necessary for all nonagenarian patients, and in our study, all patients were evaluated using DASI—in both groups, and during the second phase, with the recently established heart team in addition to the frailty score. In conclusion, due to an adequate selection of patients in the last ten years recorded in our study, the improvements obtained could be regarded as adequate preoperative care, targeting biological and nutritional levels [28], which along with the advancement of anesthesiological techniques and new methods of extracorporeal circulation, and myocardial and multi-organ protection, minimizes the biological impact.

Regarding support with the extracorporeal circulation, minimally invasive extracorporeal circulation with a closed circuit was not incorporated for any of the patients in the study, in either the first or second group, but over the years there have been technological improvements in these circuits, in terms of biocompatibility, particularly in reducing the priming volume, significantly reducing the hemodilution and consequent risks on the coagulation system.

This can also be supported because, in our study, all patients came from hospitals within the same hospital group. Different hospitals have different approaches towards patients and varying management reports, however, when compared, we recorded good results “on average”. Illustrating this, a center that operated on 40 of the 101 patients in the second group (09-21) had a 30-day mortality rate of 5%; another center that operated on eight patients had a mortality rate of 25%, which becomes 50% when electively operated patients are included. The latter further supports the evidence that it is not only a case of complications in the event of urgent/emergency treatment but also in the event of elective therapy not being managed adequately as this specific category of patients often needs an operational unit specifically for the nonagenarian type. In fact, with adequate patient selection, other studies have also been able to demonstrate that 90-year-old patients have hospital mortality outcomes comparable to 80-year-old patients [4].

Our study certainly has its limitations, the most evident of which is represented by the design of the study itself, which involves the use of a historical comparison group. Nevertheless, we want to reiterate that the aspect of the total sample size exceeding 200 nonagenarian patients is taken into account, as this number is comparatively higher than in previous studies with patients from a single hospital group. Furthermore, the two groups are not similar in preoperative characteristics or in the types of interventions received. In the more recent group, the combined interventions occur more often in percentage terms. Furthermore, regardless of more complex interventions and greater risk factors, the results were better in this group, leading to a confirmation of our hypothesis that the more recently adopted “protective” strategies are effective, even without statistical artifacts to increase the similarities between the two groups, and not making any statistical artifice necessary to make the two most “similar” groups. In other words, newer patients do better despite having a higher baseline risk. This statement also refers to the fact that the Euroscore system is not fully able to discriminate the risk in this particular category of patients.

In conclusion, our study has demonstrated that the continuous activity of evaluation, pre-, intra-, and post-operative management specific for this category of patients, and the use of strategies aimed at minimizing the biological impact on the patient, can allow an outcome in terms of 30-day mortality in nonagenarian patients that is significantly better than that recorded only a few years ago. Finally, also for the aortic valve stenosis patients, in anatomically particular situations with a contraindication for TAVI, the nonagenarian patients can undergo surgery following an accurate “heart team” evaluation.

## Figures and Tables

**Table 1 jcm-12-02177-t001:** Preoperative Characteristics.

Variable	Group 98-08	Group 09-21	*p*
	*n* = 127	*n* = 101	
Mean Age	92 ± 0.8	93.2 ± 2.5	0.16
Sex, male	62 (48.8)	45 (44.5)	0.09
NYHA functional class III or IV	58 (45.7)	39 (38.6)	0.04
Recent (<90 d) Q-wave MI	20 (15.7)	26 (25.7)	0.02
Previous cardiac surgery	2 (1.6)	2 (2)	0.18
Previous PTCA/coronary stent	5 (3.9)	8 (7.9)	0.05
Diabetes mellitus	22 (17.3)	26 (25.7)	0.03
Dyslipidemia	44 (34.6)	55 (54.4)	0.035
Systemic hypertension	81 (63.8)	90 (89.1)	0.025
Chronic renal insufficiency	12 (9.4)	24 (23.8)	0.001
Chronic obstructive pulmonary disease	14 (11)	30 (29.7)	0.001
Peripheral vasculopathy	26 (20.5)	38 (37.6)	0.06
Previous cerebrovascular accident	9 (7.1)	12 (11.9)	0.09
LVEF < 30%	5 (3.9)	30 (29)	0.01
Mean EuroSCORE (logistic)	21.3 ± 6.1	20.9 ± 11.1	0.12

MI indicates myocardial infarction; PTCA, percutaneous transluminal coronary angioplasty; and LVEF, left ventricular ejection fraction.

**Table 2 jcm-12-02177-t002:** Intraoperative Characteristics.

Variable	Group 98-08	Group 09-21	*p*
	*n* = 127	*n* = 101	
Cardiopulmonary bypass time, min	88.7 ± 32	85 ± 26	0.09
Aortic cross-clamp time, min	50.1 ± 24	43 ± 19	0.06
Elective surgical indication	82(64.6)	81 (80.2)	0.03
Isolated valve surgery	59 (46.5)	52 (52)	0.01
Isolated coronary surgery	49 (38.5)	13 (13)	
Other/combined surgery	19 (15)	36 (35)	
	Subgroup isolated AVR 98-08	Subgroup isolated AVR 09-21	
	*n* = 57	*n* = 52	
Cardiopulmonary bypass time, min	84 ± 28	87 ± 33	0.08
Aortic cross-clamp time, min	46 ± 28	51 ± 28	0.06
Minimally invasive procedure	0 (0)	18 (34.6)	<0.001
	Subgroup isolated CABG 98-08	Subgroup isolated CABG 09-21	
	*n* = 49	*n* = 13	
Cardiopulmonary bypass time, min	95 ± 42	79 ± 40	0.05
Aortic cross-clamp time, min	54 ± 22	36 ± 12	0.032
Off-pump surgery	2 (4)	10 (77)	0.03
N. of anastomosis	2.4 ± 1.2	1.3 ± 0.6	0.001

**Table 3 jcm-12-02177-t003:** Postoperative Results.

Variable	Group 98-08	Group 09-21	*p*
	*n* = 127	*n* = 101	
30-day mortality, No. (%)	17 (13.4)	6 (5.9)	0.001
ICU length of stay, d	10.2 ± 4.1	9.8 ± 5.5	0.11
Respiratory insufficiency	22 (17)	20 (19.8)	0.33
Renal failure	28 (22)	18 (17.7)	0.088
Neurological complications	14 (11)	8 (7.9)	0.08
Arrhythmias	35 (27.5)	21 (20)	0.15
Revision for bleeding	5 (3.9)	3 (2.9)	0.23
Wound infection	3 (2.4)	5 (4.9)	0.097
	Subgroup isolated AVR 98-08	Subgroup isolated AVR 09-21	*p*
	*n* = 57	*n* = 52	
30-day mortality, No. (%)	7 (12.3)	0 (0)	<0.001
ICU length of stay, d	8.8 ± 3	7.9 ± 3.3	0.07
Respiratory insufficiency	9 (15.8)	9 (17.3)	0.15
Renal failure	13 (22.8)	9 (17.3)	0.19
Neurological complications	6 (10.5)	5 (9.6)	0.32
Arrhythmias	19 (33.3)	12 (23)	0.066
Revision for bleeding	2 (3.5)	0 (0)	0.14
Wound infection	0	1 (1.9)	0.18
	Subgroup isolated CABG 98-08	Subgroup isolated CABG 09-21	*p*
	*n* = 49	*n* = 13	
30-day mortality, No. (%)	6 (12.2)	0 (0)	<0.001
ICU length of stay, d	8 ± 2.7	6 ± 1.8	0.06
Respiratory insufficiency	6 (12.2)	1 (7.7)	0.09
Renal failure	6 (12.2)	1 (7.7)	0.09
Neurological complications	3 (6.1)	1 (7.7)	0.46
Arrhythmias	13 (26.5)	2 (15.4)	0.11
Revision for bleeding	1 (2)	0 (0)	0.70
Wound infection	1 (2)	0 (0)	0.70

## Data Availability

Data sharing not applicable.

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
