# Peer review of "Cardiac Surgery in Nonagenarians Following the TAVI/TMVI Era: A Multicenter 23-Year Comparative Analysis†"

_jcm, 2023, doi:10.3390/jcm12062177_

Round 1

Reviewer 1 Report

The article by G. Nasso et al. entitled “Cardiac surgery in nonagenarians following the TAVI/TMVI era: a multicenter 23 years’ comparative analysis.” aimed to compare the short-time outcome of patients aged >89 years undergoing cardiac surgery (isolated AVR, isolated CABG and combined surgery). This is a retrospective multicenter observational study with the small sample size. The authors compared two sets of patients - patients operated from 1998 to 2008 and patients operated from 2009 to 2021. The authors found that the postoperative outcome in terms of 30-day mortality has improved significantly. The manuscript is interesting, the aim of the study is stated clearly and the introduction gives enough background. However, the inappropriate statistical methods were used in the study and statistical analysis needs to be corrected. For this reason, it is not possible to comment on the results and conclusions.

Main concern:

Has the normality of variables been checked ?

Are the assumptions of the Student's t-test met ?

Author Response

Dear Reviewer, thank you for your comments,

we added the requested information in the paper in particulare from your suggestions: we added the check for determining normality and the respect of the independence, normality and homogeneity of variances for the T-Test.

Reviewer 2 Report

The authors conducted a retrospective observational study of cardiac surgery in nonagenarians. Patients aged 90 years or over, who were operated on from 1998 to 2008 and operated on from 2009 to 2021 at 8 cardiac surgery centers, were analyzed to compare the operative characteristics, mortality and comorbidities. The outcome presented that the 09-21 group, despite having more risk factors, has a satisfactory surgical outcome and relatively low mortality. Given the sparseness of 90-year-old cases who received cardiac procedure, the present article provided some valuable information of surgical risk and perioperative management.

I have some suggestions to improve this manuscript :

(1)    In the subgroups of isolated CABG, fewer number of anastomosis was recorded in the 09-21 group. Is it due to the higher rate of off-pump CABG in this group?

(2)    Given the fact that TAVR has excellent outcome in the population of nonagenarians in the near decade, I am surprise to find that in the 09-21 group 52 patients old underwent surgical AVR. Please provide the decision-making factors of surgical AVR in this subgroup and add some comments in the Discussion section.

(3)    In the Discussion the authors mentioned the frailty has a nonnegligible effect on the surgical risk in this population. Can the authors provide more information of frailty of the study cohort?

Author Response

Thak you very much for your comments and suggestions, in particular:

1) Number of bypass was related to a higher percentage of hybrid approchaes

2) there are conditions, as also highlighted in the aforementioned guidelines18 , that can occur frequently in very elderly patients, such as a "challenging or impossible transfemoral access" or a "thrombus in aorta or left ventricle", which indicates for a conventional surgical approach rather than TAVI, regardless of age

3) the Frailty: Duke Activity Status Index (DASI) and the experience and “eye ball test” of the Heart team

Round 2

Reviewer 1 Report

I strongly recommend using an appropriate statistical method instead of "visual check". Otherwise, Authors should consider using non-parametric equivalent of the Student test. 

Author Response

Dear reviewer, we agree and we decided to check the normality of the variables using the Shapiro-Walk test. Thank you, for your input to improve the paper